# Protective Effects of Piperine on Ethanol-Induced Gastric Mucosa Injury by Oxidative Stress Inhibition

**DOI:** 10.3390/nu14224744

**Published:** 2022-11-10

**Authors:** Zhouwei Duan, Shasha Yu, Shiping Wang, Hao Deng, Lijun Guo, Hong Yang, Hui Xie

**Affiliations:** 1Institute of Agro-Products Processing and Design, Hainan Academy of Agricultural Science, Haikou 571100, China; 2Collage of Food Science and Technology, Huazhong Agricultural University, Wuhan 430070, China; 3Sanya Institute, Hainan Academy of Agricultural Sciences, Sanya 572000, China; 4College of Food Science and Technology, Hainan University, Haikou 570228, China

**Keywords:** piperine, GES-1 cells, gastric mucosal, oxidation

## Abstract

*Piper nigrum* Linnaeus is often used as a treatment for chills, stomach diseases, and other ailments. Piperine has many biological functions; however, its mechanism for preventing gastric mucosal damage is still unclear. The objective of this study was to investigate the preventive effects of piperine on ethanol-induced gastric mucosal injury by using GES-1 cells and rats. SOD, CAT, GSH-Px and MDA were effectively regulated in GES-1 cells pre-treated with piperine. Piperine significantly increased SOD, CAT and GSH-Px activities, but decreased the ulcer area, MDA, ROS and MPO levels in the gastric tissues of rats. RT-PCR analysis showed that piperine downregulated the mRNA expression levels of keap1, JNK, ERK and p38, and upregulated the mRNA transcription levels of Nrf2 and HO-1. Western blotting results indicated that piperine could activate the protein expression levels of Nrf2 and HO-1 and inhibit the protein expression levels of keap1, *p*-JNK, *p*-ERK and *p*-p38. In conclusion, piperine suppressed ethanol-induced gastric ulcers in vitro and in vivo via oxidation inhibition and improving gastric-protecting activity by regulating the Nrf2/HO-1 and MAPK signalling pathways.

## 1. Introduction

*Piper nigrum* is known as a traditional Chinese medicine for treating the stomach, indigestion, chills, muscular aches and earache [1,2]. *P. nigrum* is rich in volatile oil, alkaloids, polyphenols, starch and terpenes [3,4]. With a major pungent alkaloid, piperine is the main active ingredient of *P. nigrum*. Moreover, piperine has been proven to have many biological functions, including antioxidant, immunomodulatory, anticonvulsant, antitumor and hypolipidaemic properties [5,6,7,8,9,10]. Even though piperine might offer effective gastrointestinal protection, it is necessary to demonstrate such protection for developing and utilizing piperine.

Gastric ulcer, as an ordinary and regularly arising disease worldwide, affects almost every individual at least once in their lifetime [11]. It can be caused by many factors, including Helicobacter pylori, stress and alcohol abuse [12,13]. In daily life, gastric mucosal damage is common due to alcohol abuse. It has been widely studied for the mechanism of alcohol damage to gastric mucosa [14]. The key means of protecting gastric mucosa are the inhibition of gastric acid and anti-Helicobacter pylori treatment with drugs such as omeprazole, lansoprazole, rabeprazole and gastric mucosal protectants [15]. However, these medicines create various by-effects, including allergic reactions, haematopoietic hazards and heart abnormal beating rhythm [16]. Thus, it is essential to find natural active ingredients from raw food materials that are highly secure, have minimal by-effects and are comfortable in order to protect the stomach.

Ethanol is a substance that causes mucosal necrosis. When it acts on gastric mucosal epithelial cells for a long time, ethanol can corrode the mucus layer of epithelial cells, stimulate gastric secretion and aggravate the damage of gastric mucosa; on the other hand, ethanol directly acting on gastric mucosal cells will lead to cell dehydration, degeneration and necrosis [17]. It is common to use ethanol-induced human gastric epithelium (GES-1) cells and gastric ulcer rat models for studying acute gastritis [18]. When the internal environment of the gastric mucosa is under oxidative pressure because of alcohol stimulation, it decreases the production of endogenous glutathione and prostaglandins and increases the release of histamine, free radicals and leukotrienes on the basis of mucosal barrier damage, resulting in further gastric damage to the mucosa [19]. Many studies have revealed that there is uniform physiologic and morphological variability in the haemorrhagic, mucus and gastric acid of rats as in humans [20]. Excessive reactive oxygen species (ROS) stimulate oxidative damage and inflammatory responses in gastric mucosa and are tightly linked to the formation of gastric damage [19,21].

Studies have demonstrated that piperine exhibits various biological functions, including antioxidant, immunomodulatory and anti-inflammatory activities [5,6,7,8,9,10]. Therefore, it is reasonable to hypothesize that piperine could be probably advantageous for relieving gastric mucosal harm caused by ethanol activation. Hence, this research aims to study the gastroprotective effect from piperine upon GES-1 cells and rats with ethanol-stimulated gastric ulcers. The results achieved from this research could possibly offer a scientific foundation for applying piperine as a natural product to protect gastric tissue, and to be used as a prospective raw material for functional foodstuffs.

## 2. Material and Methods

### 2.1. Materials and Chemicals

White pepper was obtained from Qionghai Hongsheng Food Co., Ltd. (Hainan, China). High-dextrose DEME and Fetal bovine serum (FBS) were purchased from Gibco Life Technologies (Waltham, MA, USA). β-actin, Nrf2, JNK, keap1, HO-1, ERK, p38, *p*-JNK, *p*-ERK and *p*-p38 antibodies were from Abcam (Cambridgeshire, UK). Superoxide dismutase (SOD), catalase (CAT), glutathione peroxidase (GSH-Px) and malondialdehyde (MDA) were purchased from Nanjing Jiancheng Bioengineering Institute (Nanjing, China). Reactive oxygen species (ROS) and myeloperoxidase (MPO) ELISA Kits were obtained from Shanghai XinYu Bio-Technology Co., Ltd. (Shanghai, China). Lansoprazole (LAN) was purchased from Beijing Soleibo Technology Co., Ltd. (Beijing, China). RNA extraction kit was from Tiangen Biotech (Beijing) Co., Ltd. (Beijing, China). The primers of Nrf2, JNK, keap1, HO-1, ERK, p38 and GADPH were synthesized by Sheng Gong Co., Ltd. (Shanghai, China).

### 2.2. Preparation of Piperine

Piperine extracted and purified from *P. nigrum* followed our previous research [22].

### 2.3. Cell Experiment

#### 2.3.1. Cell Culture Conditions

GES-1 cells, obtained from BeNa Culture Collection (Xinyang, China), were maintained as the methods [18].

#### 2.3.2. Piperine Cytotoxicity Analysis

The cytotoxic effects of piperine were evaluated by the MTT method [18]. GES-1 cells were seeded in 96-well plates (100 μL/well) at a density of 1 × 10^4^ cells/mL for 24 h. The medium was substituted with a new medium including different concentrations of piperine (from 5 to 120 mg/L), while new medium was added to the control group. After culturing for 24 h, 10 μL of 5 mg/mL MTT solution was dissolved in medium and a 4-h incubation continued in the darkness. Afterwards, the medium was discarded and the formazans were dissolved in DMSO (100 μL). The absorbance of the cell suspension was examined at 490 nm using a microplate reader (ThermoFisher, Waltham, MA, USA).

#### 2.3.3. Protective Effect of Piperine

The procedure of cell incubation was the same as that described in Section 2.3.2. The cells were separated into control, ethanol and experimental groups. Then the experimental group was pre-handled with 5, 10 and 20 mg/L concentrations of piperine, while new medium was put into the other two groups. After incubation for 24 h, the supernatant was discarded. Ethanol solution was added to the ethanol and experimental groups at a final concentration of 1.0 mol/L and incubated for 2 h. Finally, cell viability was evaluated by the MTT method [18].

#### 2.3.4. SOD, CAT, GSH-Px and MDA Assays

The procedure to culture and group cells was as described in Section 2.3.3, except cells were plated in 12-well plates at 500 μL/well. After incubating for 2 h, the supernatant was removed. The activities of SOD, CAT, GSH-Px and the level of MDA were quantified by assay kits.

### 2.4. Rat Experiment

#### 2.4.1. Rat Treatment

Female Sprague-Dawley rats (200 ± 10 g) were supplied by the TianQin Biotechnology (Changsha) Company in Changsha, China, with a licensed ID: SCXK 2019-0013. Animal procedures were approved by the Hainan University Institutional Animal Use and Care Committee (HNDX2020072). The rats were acclimatized to a temperature of 23 ± 1 °C (relative humidity of 65–70%) with a cycle, light and dark for 12 h, and provided with food and water as much as necessary.

After the rats have been adjusted for 7 days, six groups were formed from 72 rats randomly, each group having 12. The groups were as follows: normal group (NC), model group (MC), positive group (LAN, 30 mg/kg. bw, lansoprazole), low-dose group (PL, 25 mg/kg. bw, piperine), middle-dose group (PM, 50 mg/kg bw, piperine) and high-dose group (PH, 100 mg/kg bw, piperine). Rats from the NC group and the MC group were supplied with 5% Tween 80 distilled in water for gavage daily. Rats in the LAN, PL, PM and PH groups were administered LAN or the corresponding doses of piperine in a daily oral aqueous solution. After 7 days of pretreatment, all rats were kept fasting but watered for 24 h. The anterior abdominal wall from the xiphoid process to the anus was incised along the midline of the abdomen, and the stomach was removed. Except for the NC group, the others were given 4 mL/kg absolute ethanol to make the model. After treatment for 4 h, blood was collected. The anterior abdominal wall from the xiphoid process to the anus was incised along the midline of the abdomen, and the stomach was removed.

#### 2.4.2. Specimen Collection and Preparation

The collection of blood and stomach tissue followed previous research [23]. The ulcer area and ulceration protection were calculated by these methods [24,25].

#### 2.4.3. Histopathological Assessment

The section of gastric injury was assessed by haematoxylin and eosin (H&E), which was performed using light microscopy with 200× and 400× magnification power [23].

#### 2.4.4. Evaluation of SOD, CAT, GSH-Px, MDA, ROS and MPO

Gastric tissues were mixed with cold phosphate buffered saline to prepare 10% homogenate, which was centrifuged at 4000× *g* rpm at 4 °C for 10 min. The levels of SOD, CAT, GSH-Px and MDA were quantified by biochemical kits. The levels of ROS, and MPO in the gastric tissues, were determined following ELISA kit instructions [26].

#### 2.4.5. RT-qPCR Analysis

Total RNA was collected from gastric tissue following the instructions of the RNA extraction kit, and then 0.5 μg RNA was applied for reverse transcription under the following conditions, with a temperature 42 °C for 15 min and 95 °C for 3 min. Hereafter, cDNA was amplified by quantitative RT-PCR using the SuperReal RreMIx Plus Kit. Next, RNA was transcribed into cDNA as follows, 95 °C for 15 min, 40 cycles of 95 °C for 10 s and 60 °C for 30 s. The total volume of reverse transcriptions for RNA was measured in line with the concentration. The primer sequences of the genes are displayed in Table 1. The level data gene transcription was computed in a method of 2^−ΔΔCt^ [27,28].

#### 2.4.6. Western Blot Analysis

After total proteins have been separated from gastric tissues, the quantification was performed by a bicinchoninic acid (BCA) kit. The total proteins were detached by SDS-PAGE and transferred to polyvinylidene difluoride (PVDF) membranes. Later, Western blot analysis was measured as in the reports [29,30].

### 2.5. Statistical Analysis

All values are presented as the means ± standard deviations (SD). Significant difference analysis was performed by one-way analysis of variance (ANOVA) with Duncan’s test. *p* < 0.05 was considered statistically significant.

## 3. Results

### 3.1. Proliferation of Piperine on GES-1 Cells

Prior to application in cell tests, natural active compounds usually need a virulence experiment to assess their safety [31]. When piperine had lower concentration than 40 mg/L, piperine had no significant effect on cell viability (Figure 1A). When the concentration of piperine was up to 60 mg/L, the cell viability dropped to 95.82 ± 0.27%, which was considerably lower than that of the control group (*p* < 0.05). After handling of 80 mg/L piperine, cell viability (93.11 ± 1.47%) was significantly lower than that of the control (*p* < 0.01). The results indicated that piperine displayed no cytotoxicity in GES-1 cells when concentrations were below 40 mg/L. Combined with the pre-experimental results, the concentrations of piperine at 5, 10 and 20 mg/L were suitable for experiments in the future.

In the ethanol group, the cell viability of GES-1 cells was 47.11 ± 1.01% after induction with ethanol for 2 h. The data suggest that the ethanol-induced GES-1 cell model was successfully established. When treated with piperine at 5–20 mg/L, the cell viabilities were significantly increased and were higher than those of the ethanol group (*p* < 0.05). In particular, after handling of 20 mg/L piperine, the cell viability increased to 68.01 ± 1.30% (Figure 1B). This indicates that the cell protective effect of piperine was concentration dependent.

### 3.2. Effects of Piperine on the Antioxidant Parameters

The antioxidant indexes, such as activities of SOD, CAT and GSH-Px, and levels ofMDA, were considered to analyze the varieties in the oxidative stress of cells. As seen in Figure 2A–D for the ethanol-induced group, the enzymatic activities of SOD, CAT and GSH-Px were dramatically reduced (*p* < 0.01), which revealed that the antioxidative capacity was restrained; the levels of MDA were significantly increased (*p* < 0.01), which indicated increased oxidative stress damage. However, compared with ethanol treatment, piperine (5–20 mg/L) treatment significantly enhanced the enzyme activities of SOD, CAT and GSH-Px and reduced MDA levels in accordance with dosage (*p* < 0.01). Interestingly, in accordance with the enzyme activity of GSH-Px, there was little disparity between the high-dose piperine (20 mg/L) group and the control group. Piperine possibly increased the enzyme activity of SOD, CAT and GSH-Px and lowered MDA levels in GES-1 cells injured by ethanol.

### 3.3. Preventive Effect of Piperine on Ethanol-Stimulated Gastric Ulcers in Rats

Figure 3 shows macroscopic data for gastric tissue. The NC group displays the surface of the gastric tissue complete, sleek and natural. No harm, edema or congestion was observed. However, in the MC group, there was spot-like or strip-like congestion on the mucosal surface with edema, large injury area, serious injury and erosion. No congestion or edema was found in the LAN group, which was close to the NC group. The PL group had punctate and thread-like haemorrhage, and the injury was not as severe as that from the MC group, but more severe than that from the LAN group. The PM group had lighter bleeding than the PL group, with only a few spots or short band haemorrhages. The PH group was basically free of congestion and edema, which was close to the NC and LAN groups (Figure 3A).

From the H&E-stained pathological sections in Figure 3B,C, it could be seen that the rats’ gastric tissue in the NC group had a clear cell structure, intact glandular structure, compact and orderly arrangement of cells, and no bleeding or inflammatory cells. The rats’ gastric mucosa in the MC group had obvious bleeding, gastric gland epithelial shedding, gland structure disappearance, ulcer or perforation and other phenomena. In contrast, the epithelial cells were partially shed for the rats pretreated with LAN and piperine, while the mucosal hemorrhage range was reduced and the cell arrangement tended to be compact and orderly. Among them, the degree of improvement in the PH group was, importantly, stronger than that in the PM and PL groups.

As shown in Figure 3D,E, in the NC group, the ulcer area and the protection of ulceration in gastric mucosa were 0 and 100%, respectively. In contrast to the MC group, the LAN, PL, PM and PH groups decreased the ulcer area of the gastric mucosa and improved the protection of ulceration. The ulcer areas of the MC, LAN, PL, PM, and PH groups were 52.91 ± 4.46, 4.44 ± 0.21, 45.12 ± 4.33, 16.07 ± 1.75, and 3.87 ± 0.23 mm^2^, respectively; the protection of ulceration was 0%, 92.68 ± 0.27%, 13.03 ± 2.86%, 62.61 ± 6.29%, and 93.38 ± 2.21%, respectively. The LAN, PL, PM and PH groups had dramatic differences from the MC group (*p* < 0.01). The results showed that piperine could effectively protect gastric mucosal from injury stimulated by ethanol. In particular, the rats pretreated with piperine (100 mg/L) and LAN (30 mg/L) presented equivalent protection against gastric mucosal injury (Figure 3A,D,E).

### 3.4. Evaluation of SOD, CAT, GSH-Px, MDA, ROS and MPO

The results for gastric tissue from biochemistry indices are shown in Figure 4. In contrast to the MC group, the levels of MDA, ROS and MPO in the LAN, PL, PM and PH groups decreased dramatically, while the activity of SOD, CAT and GSH-Px increased and was associated with enhancing piperine doses (Figure 4A–F). In contrast to the MC group, the activity of SOD, CAT and GSH-Px in the LAN, PL, PM and PH groups increased 74.36%, 95.07%, 103.26% and 124.46%; 24.99%, 14.90%, 22.64% and 36.97%; 40.14%, 16.10%, 31.84% and 34.71%, respectively (Figure 4A–C). In Figure 4D, the dilution levels of MDA in the LAN group and piperine groups were decreased dramatically in comparison with the MC group. There were dramatically significant decreases in the levels of ROS and MPO in the LAN and piperine groups with three concentrations compared with the MC group (Figure 4E,F). Apparently, piperine has the effect of decreasing the levels of MDA, ROS and MPO and improving the activity of SOD, CAT and GSH-Px in gastric tissues.

### 3.5. mRNA Expression Levels of Oxidation-Related Genes

The indications of oxidation-connected genes (Nrf2, HO-1, Keap1, ERK, JNK and p38) were ascertained by RT-PCR to determine the protective effects on gastric tissues from piperine. In contrast to the NC group, the mRNA indications of Nrf2 were dramatically decreased in the MC group, while HO-1 had no significant differences (Figure 5A,C). Compared with the MC group, the mRNA expression levels of Nrf2 and HO-1 were improved in the rats’ gastric tissues, pre-handled with LAN or piperine. Treated with 20 mg/L piperine, the transcription levels of Nrf2 and HO-1 mRNA were improved by 369.62% and 143.16%, respectively. However, the mRNA expression levels of Keap1, ERK, JNK and p38 were tremendously augmented in the rats´ gastric tissues, pretreated with LAN and all three kinds of doses from piperine in comparison with the MC groups. Moreover, in contrast to the MC group, the mRNA indications of Keap1, ERK, JNK and p38 in the PH groups were reduced by 56.79%, 51.30%, 44.81% and 43.86%, respectively (Figure 5B,D–F). Notably, piperine has significant effects in reducing oxidation established by improving the mRNA expression levels of Nrf2 and HO-1 and decreasing the mRNA transcription levels of Keap1, ERK, JNK and p38, and the PH group showed an even better effect.

### 3.6. Analysis of Nrf2, HO-1, Keap1, ERK, JNK and p38 Expression

To further pursue our research, the protein indications of Nrf2, HO-1, Keap1, ERK, JNK and p38 were measured by Western blotting. The protein expressions of Nrf2, Keap1, *p*-ERK, *p*-JNK and *p*-p38 were significantly different between the NC and MC groups, while HO-1 had no significant differences. Ethanol intubation significantly downregulated Nrf2 and enhanced Keap1, *p*-ERK, *p*-JNK and *p*-p38 relative to the NC group (Figure 6). In addition, the rats pretreated with LAN or piperine exhibited significantly improved protein of Nrf2 and HO-1 and reduced protein indications of Keap1, *p*-ERK, *p*-JNK and *p*-p38 (Figure 6B–G). Clearly, in contrast to the MC group, the PH group illustrated the optimal influence on increasing the protein indications of Nrf2 and HO-1 by 148.87% and 71.65%, respectively, and reducing the protein expression of Keap1, *p*-ERK, *p*-JNK, *p*-p38 by 68.21%, 40.10%, 82.13% and 82.87%, respectively (Figure 6B–G). The results revealed that piperine has dramatic effects on regulating the protein indications of Nrf2, HO-1, Keap1, ERK, JNK and p38.

## 4. Discussion

*P. nigrum* is used to treat many illnesses, such as stomach disorders, in Chinese medicine. Researchers have demonstrated that piperine possesses antioxidant, anti-inflammatory and immune enhancement activities, indicating its underlying antiulcer activity. According to our previous studies, there was 90.65 ± 0.46% purity for the piperine extracted and purified from *P. nigrum* [22]. Therefore, we conducted research to study the possibilities to fight against ulcer and protect gastric tissues with piperine and to evaluate the potential function of how it works.

The ethanol-induced GES-1 cell model is frequently applied to evaluate an agent’s activity against ulcer [18]. SOD is an endogenous antioxidant enzyme for the body that can remove excessive ROS and MDA in the body. It protects the body and plays an important role in the balance of cellular oxidation and antioxidants [32]. CAT is an essential enzyme in cells that can effectively remove various active oxygen groups and prevent them from damaging the cell membrane system [33]. GSH-Px can catalyze the conversion of GSH into oxidized glutathione and reduce peroxides to hydroxyl compounds. GSH-Px and CAT contribute to the clearance of free radicals to protect the structure and function of cell membranes [34]. MDA is an important metabolite generated after the decomposition of peroxidation, and its content reveals the degree of free radical injury to the cells [32]. In this experiment, it was indicated that ethanol stimulates peroxidation in GES-1 cells via pathways including the reduction of SOD, CAT, GSH-Px and advances in the levels of MDA. The results revealed that piperine could increase the activities of intracellular antioxidant enzymes and prevent peroxidation to defend cell membranes (Figure 2A–D).

Ethanol has a dehydrating effect, and when ethanol is administered in the stomach tissue, it can corrode the gastric mucosa, which can result in gastric mucosal congestion, edema, erosion, ulceration or bleeding and cause damage to the gastric mucosal cell membrane [35]. When the concentration or intake of ethanol exceeds the acceptable range of the gastric mucosa, it changes the balance of the gastric mucosal barrier, stimulates the infiltration and release of inflammatory cells and inflammatory factors, and changes the microcirculation state of the gastric mucosa, eventually triggering the offspring of oxygen free radicals and inducing mucosal lipid peroxidation, inflammatory cell infiltration and other phenomena [36,37]. The ethanol-stimulated gastric ulcer in rats expresses numerous aspects similar to gastric ulcer in human beings, and therefore it is normally used in evaluating the activity of an antiulcer effect; likewise, the possible pathways take part in this process [38,39,40]. After 7 days of pretreatment with piperine (25, 50 and 100 mg/kg, bw), the ethanol (4 mL/kg)-stimulated gastric ulcer model led to serious oxidative-stress-induced injury in rats (Figure 3), and the results were consistent with previous research [41]. Interestingly, pretreatment with piperine (100 mg/kg, bw) resulted in 93.38 ± 2.21% gastroprotection, which was similar to that of the LAN group (Figure 3E). This impact on anti-ulcer effect was showed by histological examination exhibiting complete mucosal conformation and glandular substances, an abatement in the submucosal edema and inflammatory cell permeation (Figure 3B,C). In addition, the gastroprotective function from piperine was associated with the improvement of mucosal membrane protection. These results suggest that piperine can effectively prevent gastric mucosal injury induced by ethanol, which is in agreement with the results of piperine protecting GES-1 cells.

Oxidative stress has been closely related to gastric mucosal injury. Gastric mucosa injury can be lightly generated by the production of detrimental free radicals, and imbalances or declines in gastric mucosa antioxidant defence systems have been part of the pathogenesis and evolution of gastric ulcers [42,43]. Superfluous ethanol produces injurious ROS and free radicals and participates in the etiopathogenesis of gastric mucosa [44]. As a biological symbol for neutrophil percolation, the enzyme MPO acts as a vital part in the continuum of damage in numerous tissues [45]. In this experiment, ethanol alone significantly increased the MDA, ROS and MPO levels and decreased SOD, CAT and GSH-Px activities in gastric tissue, while pretreatment with piperine significantly reduced MDA, ROS and MPO levels and improved serum SOD, CAT and GSH-Px activities (Figure 4A–F), which was consistent with previous studies [23]. The fabrication of ROS by ethanol is able to cause injury to the stomach, while enhanced antioxidative enzymes can defend gastric mucosa from anabrosis [46]. In particular, the activity of SOD, CAT and GSH-Px and the level of MDA in gastric tissue and GES-1 cells showed a similar change trend. The results suggest that piperine improves the activities of antioxidant enzymes and scavenges intracellular ROS to defend rats´ gastric mucosa.

To investigate the signalling pathway of piperine, the gene and protein indications of Nrf2, HO-1 and Keap1 were determined. The Nrf2/HO-1 signalling pathway is considered an important signal in defending cells from oxidative-stress-induced destruction [47,48]. The RT-PCR results showed that piperine (25–100 mg/kg, bw) dramatically enhanced the mRNA indications of Nrf2 and HO-1 and reduced the secretion of keap-1 mRNA (Figure 5A–C). These results were similar to Yanaka’s research [49]. In the oxidative–antioxidative imbalance in the body, the cysteine residue of Keap1 is modified, changing its conformation, advancing Nrf2 movement from the cytoplasm to the nucleus, forming a heterodimer with the small Maf protein, and binding to the ARE to induce downstream expression of HO-1, CAT and other antioxidant enzymes [49]. HO-1 is one of the most gastrointestinal protective enzymes, which can enhance cytoprotection, inhibit oxidative stress and depress apoptosis by restraining ROS [21,50]. In the present study, ethanol alone significantly increased the keap-1 level and decreased Nrf2 level in gastric tissue. Significantly, LAN and piperine significantly reduced keap-1 levels and improved Nrf2 level and HO-1 activity, suggesting Nrf2/HO-1 signalling activation (Figure 4A–D). Piperine regulates the Nrf2/HO-1 anti-oxidative pathway to protect the gastric mucosa.

The MAPK signalling way is a vital path to regulate cell proliferation and apoptosis, and it participates in the regulation of abundant cell life cycles in cells [51]. MAPK contains three important signalling pathways: extracellular signal-regulated kinase (ERK), c-Jun-N-terminal kinase (JNK) and p38 [52]. When external stimuli are transmitted to cells, the MAPK signalling pathway is partially or fully activated, and each signalling pathway can be independently regulated or can coordinate to regulate cell function [53]. ERK, JNK and p38 in MAPK signalling can be activated by ethanol and produce oxygen free radicals, resulting in a series of stimulatory responses in the body [54]. Piperine has significant effects in decreasing the mRNA transcription levels of ERK, JNK and p38 (Figure 5D–F). The Western blot results suggest that piperine (25–100 mg/kg, bw) dramatically reduced the indications of *p*-ERK, *p*-JNK and *p*-p38 (Figure 6E–G).

Our findings further confirm that the piperine extract from *P. nigrum* exhibits a strong preventive capacity against ethanol-stimulated gastric mucosa damage in GES-1 cells and rats. However, the situation in ethanol-gastric ulcers is complex and is closely related to oxidative stress, inflammatory reactions and mitochondrial pathway apoptosis. Next, we will further study the preventive mechanism of piperine on ethanol-induced gastric mucosa in terms of the inflammatory response and cell apoptosis.

## 5. Conclusions

In this study, the cytotoxicity and protective influence of piperine on ethanol-stimulated GES-1 cells were investigated. The results suggest that piperine could improve the antioxidant effect between cells, adjust the cellular redox system (MDA) and increase the intracellular antioxidant enzymes (SOD, CAT, GSH-Px). In rats with ethanol-induced ulcers, piperine decreased the ulcer area, increased SOD, CAT and GSH-Px activities, and reduced MDA, ROS and MPO levels. Moreover, piperine altered the mRNA and protein expression levels of Nrf2, keap1, HO-1, JNK, ERK, and p38, which may be closely related to the regulation of the Nrf2/HO-1 and MAPK pathways. In general, piperine exhibits strong gastric mucosa protective activity in vitro and in vivo. This work offers further understanding of piperine targets and supports the development of functional foods to protect the gastric mucosa.

## Figures and Tables

**Figure 1 nutrients-14-04744-f001:**
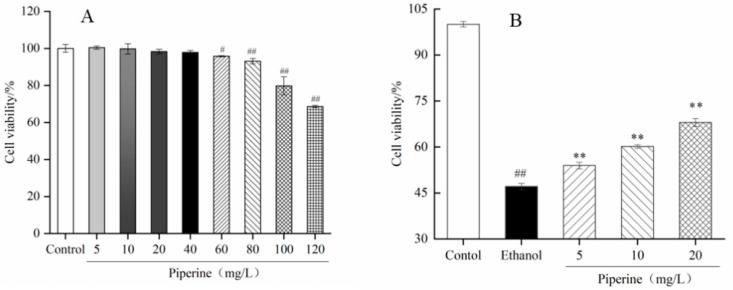
Effect of piperine on GES-1 cells. (**A**) Cell viability. (**B**) Protective activity of piperine against ethanol-induced GES-1 cells. Ave ± SD, ^#^
*p* < 0.05, ^##^
*p* < 0.01 vs. control group; ** *p* < 0.01 vs. ethanol group.

**Figure 2 nutrients-14-04744-f002:**
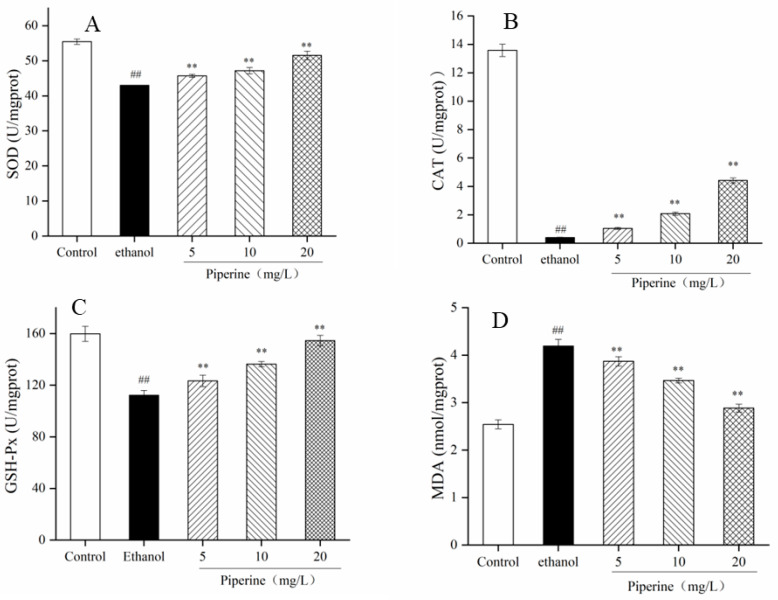
Effect of piperine pretreatment on oxidative factors in ethanol-treated GES-1 cells. (**A**) SOD; (**B**) CAT; (**C**) GSH-Px; (**D**) MDA. Ave ± SD, ^##^
*p* < 0.01 vs. control group; ** *p* < 0.01 vs. ethanol group.

**Figure 3 nutrients-14-04744-f003:**
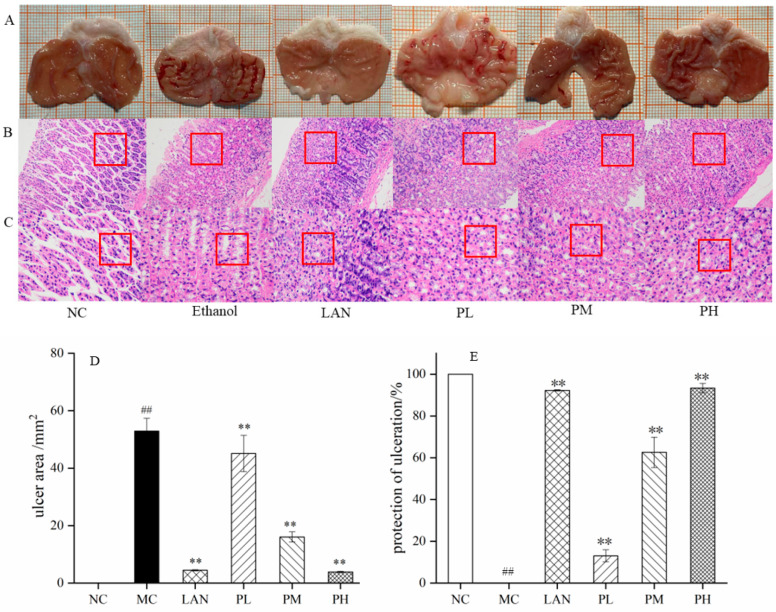
Effect of piperine on ethanol-induced gastric mucosa in rats. (**A**) Macroscopic evaluation of rat gastric tissue, (**B**,**C**) H&E staining Sections 200× and 400×, (**D**) ulcer area, and (**E**) ulceration protection. Ave ± SD, ^##^
*p* < 0.01 vs. NC group; ** *p* < 0.01 vs. MC group.

**Figure 4 nutrients-14-04744-f004:**
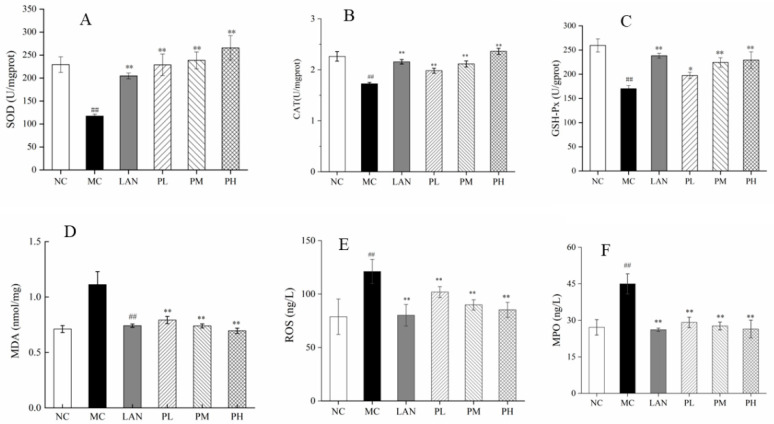
Effect of piperine pretreatment on biochemical indicators in gastric tissue. (**A**) SOD; (**B**) CAT; (**C**) GSH-Px; (**D**) MDA; (**E**) ROS; (**F**) MPO. Ave ± SD, ^##^
*p* < 0.01 vs. NC group; * *p* < 0.05 and ** *p* < 0.01 vs. MC group.

**Figure 5 nutrients-14-04744-f005:**
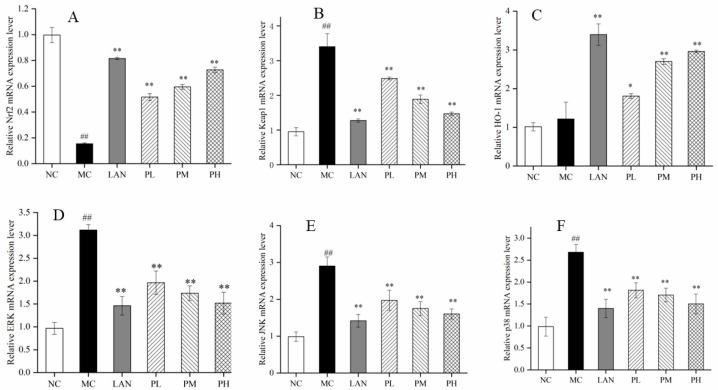
Effects of piperine on the mRNA expression of Nrf2 (**A**), Keap1 (**B**), HO-1 (**C**), ERK (**D**), JNK (**E**) and p38 (**F**) in the full-thickness gastric tissue of the rats. Ave ± SD, ^##^
*p* < 0.01 vs. NC group; * *p* < 0.05 and ** *p* < 0.01 vs. MC group.

**Figure 6 nutrients-14-04744-f006:**
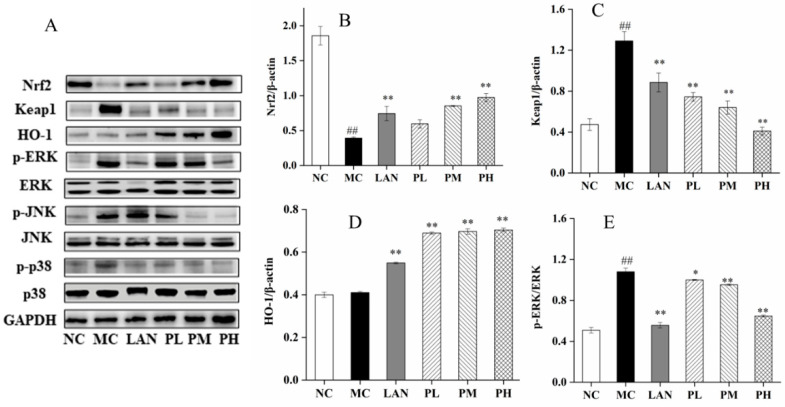
Piperine inhibited ethanol-induced oxidative effects through Nrf2/ HO-1 and MAPK pathways in the gastric tissue of the rats. (**A**) The protein levels of Nrf2, HO-1, Keap1, *p*-ERK, ERK, *p*-JNK, JNK, *p*-p38 and p38 were measured by Western blot. The relative levels of (**B**) Nrf2, (**C**) Keap1, (**D**) HO-1, (**E**) *p*-ERK/ERK, (**F**) *p*-JNK/JNK, (**G**) *p*-p38/p38. Ave ± SD, ^#^
*p* < 0.05, ^##^
*p* < 0.01 vs. NC group; * *p* < 0.05 and *** p* < 0.01 vs. MC group.

**Table 1 nutrients-14-04744-t001:** Genes and primers for PCR.

Gene	Primer Sequence (5′-3′)	Orientation	Length/bp
GADPH	GACATGCCGCCTGGAGAAAC	Forward	92
	AGCCCAGGATGCCCTTTAGT	Reverse	
Nrf2	GCCTTCCTCTGCTGCCATTAGTC	Forward	110
	TGCCTTCAGTGTGCTTCTGGTTG	Reverse	
HO-1	CAGGTGTCCAGGGAAGGCTTTAAG	Forward	96
	TGGGTTCTGCTTGTTTCGCTCTATC	Reverse	
Keap1	TGCTCAACCGCTTGCTGTATGC	Forward	99
	TCATCCGCCACTCATTCCTCTCC	Reverse	
ERK	CTGGCTGCTAGGAACATTCTGGTG	Forward	84
	GTCATCCTGGAGGTAGCGAGAGAG	Reverse	
JNK	CCACCACCAAAGATCCCTGACAAG	Forward	117
	GACGCCATTCTTAGTTCGCTCCTC	Reverse	
p38	GATAAGAGGATCACAGCAGCCCAAG	Forward	149
	TCGTAGGTCAGGCTCTTCCATTCG	Reverse	

## Data Availability

Data are contained within the article.

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
