# Peer review of "Protective Effects of Piperine on Ethanol-Induced Gastric Mucosa Injury by Oxidative Stress Inhibition"

_nutrients, 2022, doi:10.3390/nu14224744_

Round 1

Reviewer 1 Report

In the manuscript entitled Protective effects of piperine on ethanol-induced gastric mucosa injury by oxidative stress the authors presented data of comprehensive study about effects of piperine on gastric mucosa. The results presented in this scientific paper significantly enlarge current knowledge about piperine health effects. 

I have only several suggestions and comments that could, by my opinion contribute to just slightly improve the quality of this original, well written paper.

List of minor suggestions

1. Please check is MDA is maleic dialdehyde or malonyl dialdehyde

2. It could be useful to readers to add in introduction section what lansoprazole is as it is used as positive control, in a sentence or within sentence where drugs for ulcer are listed (line 41-43)

3. Some explanations why different parameters of oxidative status are determined in GES-1 cells (SOD, CAT, GSH-Px, MDA) compared to those in gastric tissue (ROS, CAT, MDA, MPO) could be of interest for researcher if they are planning to conduct similar studies, experiments

4. The kits for determination of SOD, CAT, GSH-Px, MDA, MPO are routinely used, but it may be informative to readers to briefly describe method for ROS determination.

5. The Figure 5. legend should be corrected as 5B) is not representing HO-1 expression but Keap1, and 5C) represent Keap1 expression not HO-1

6. The authors could add some additional comment about effect of piperin and lansoprazole on OH-1 expression

Best regards

Author Response

Dear reviewer:

     We are very grateful to you for the effort and time spent reviewing our manuscript with the original title ‘Protective effects of piperine on ethanol-induced gastric mucosa injury by oxidative stress inhibition’. Following the comments of you, we have modified our manuscript. Any revisions to the manuscript using the “*Track Changes*” function in the MS Word.

Point 1: Please check is MDA is maleic dialdehyde or malonyl dialdehyde

Response 1: Thank you for your suggestion. MDA is malondialdehyde in line 82.

Point 2: It could be useful to readers to add in introduction section what lansoprazole is as it is used as positive control, in a sentence or within sentence where drugs for ulcer are listed (line 41-43)

Response 2: Thank you for your suggestion. According to your suggestion, lansoprazole is as it is used as positive control in line 44.

Point 3: Some explanations why different parameters of oxidative status are determined in GES-1 cells (SOD, CAT, GSH-Px, MDA) compared to those in gastric tissue (ROS, CAT, MDA, MPO) could be of interest for researcher if they are planning to conduct similar studies, experiments

Response 3: Thank you for your suggestion. We had determined the activities of SOD, CAT and GSH-Px. According to your suggestion, we supply the data of SOD and GSH-Px in gastric tissue in section 2.4.4 and 3.4. The ROS and MPO changes are closely associated with the oxidative damage and the inflammatory response. The determination of ROS and MPO changes in gastric tissue was conducted to evaluate the effect of piperine in preventing gastric mucosal damage comprehensively.

Point 4: The kits for determination of SOD, CAT, GSH-Px, MDA, MPO are routinely used, but it may be informative to readers to briefly describe method for ROS determination.

Response 4: Thank you for your suggestion. According to your suggestion, We supply some pretreatment methods in line 151-154.

Point 5: The Figure 5. legend should be corrected as 5B) is not representing HO-1 expression but Keap1, and 5C) represent Keap1 expression not HO-1

Response 5: Thank you for your suggestion. Figure 5B and 5C is Keap1and HO-1, respectively (line 291).

Point 6: The authors could add some additional comment about effect of piperin and lansoprazole on OH-1 expression.

Response 6: Thank you for your suggestion. According to your suggestion, we add some additional comment about effect of piperine and lansoprazole on OH-1 expression in line 276, 300, 393-398.

Reviewer 2 Report

This is a nice study by Duan et al investigating the protective effects of piperine on ethanol-induced gastric mucosal injury. The authors identify a role for piperine in mitigating oxidative stress when administered to cells and rats prior to an ethanol bolus which reduces ulcer formation. 

What kind of foods include piperine? Please describe this food in layman’s terms. (line 28)

Please introduce what kind of cells are GES-1 cells – are these normal or from an ulcer or metaplastic model? (line 52)

Please include a citation for the wide ranges of functions for piperine (line 63).

Please avoid use of the term “extremely significant” – “significantly” suffices.

Line 172 – the figure demonstrates that cell viability was lower, not higher, at 80mg/l

Line 178 – please describe in the main text whether cells were treated with ethanol then piperine, or piperine then ethanol, and the duration of treatment.

Please write the full name of the antioxidants and describe their function upon first use (line 186). 

Section 3.3 – please describe your experimental groups and abbreviations in the main text. 

Figure 3B-C – it is difficult to tell from the H&E images what is defined as ulceration and what is defined as protected from ulceration, and how these images were quantified. The PH image looks nothing like the NC image, and yet the quantification indicates these are nearly the same. Please assist the reader in identifying normal mucosa versus regions of ulceration in these images. 

Figure 4 – Please describe more carefully how these samples were obtained. ROS levels change dramatically upon death of the animal and dissection of the tissue. Were these mucosal samples, full-thickness gastric tissue, or blood? These details should be presented in both the main text as well as the figure legend. 

Figure 5 – were these tissues collected from full-thickness gastric tissue or only mucosa? 

Figure 6 – were entire stomachs, including ulcerated tissues, included for protein collection? Please include this information in the methods section. 

For all in vivo studies – while the graphs depict a trend toward best results at the higher dose of piperine, is there a significant difference between the three doses tested? For translation into humans, the lowest effective dose will be important to ascertain. 

Many of the details pertinent to the study are included in the discussion when they would be better suited to the introduction (description of antioxidants and effects of ethanol on the stomach).

Line 325-328 – there was no data describing lipid peroxidation in this manuscript. Oxidative stress can occur from numerous other sources and this study did not determine the cause. 

Author Response

Dear reviewer:

We are very grateful to you for the effort and time spent reviewing our manuscript with the original title ‘Protective effects of piperine on ethanol-induced gastric mucosa injury by oxidative stress inhibition’. Following the comments of you, we have modified our manuscript. Any revisions to the manuscript using the “*Track Changes*” function in the MS Word.

Point 1: What kind of foods include piperine? Please describe this food in layman’s terms. (line 28)

Response 1: Thank you for your suggestion. Piperine mainly found in Piper nigrum L and Piper longum L. The meaning of food (in line 28) is dual-use resource of medicine and food.

Point 2:Please introduce what kind of cells are GES-1 cells – are these normal or from an ulcer or metaplastic model? (line 52)

Response 2: Thank you for your suggestion. GES-1 cells are the human gastric epithelium (normal) in line 55.

Point 3:Please include a citation for the wide ranges of functions for piperine (line 63).

Response 3: Thank you for your suggestion. According to your suggestion, we cited the wide ranges of functions for piperine in line 67.

Point 4:Please avoid use of the term “extremely significant” – “significantly” suffices.

Response 4: Thank you for your suggestion. According to your suggestion, we use the term of significantly in line 179, 188, 196, 208.

Point 5:Line 172 – the figure demonstrates that cell viability was lower, not higher, at 80mg/l

Response 5: Thank you for your suggestion. The figure demonstrates that cell viability was lower at 80 mg/l in line 188.

Point 6:Line 178 – please describe in the main text whether cells were treated with ethanol then piperine, or piperine then ethanol, and the duration of treatment.

Response 6: Thank you for your suggestion. The cells were treated with piperine then ethanol in experimental groups (Section 2.3.3). We describe the cell treating steps in the main text in line 193.

Point 7:Please write the full name of the antioxidants and describe their function upon first use (line 186).

Response 7: Thank you for your suggestion. We have write the full name of the antioxidants in line 81-84.

Point 8:Section 3.3 – please describe your experimental groups and abbreviations in the main text.

Response 8: Thank you for your suggestion. We describe experimental groups and abbreviations in in 2.4.1 (line 132-135).

Point 9:Figure 3B-C – it is difficult to tell from the H&E images what is defined as ulceration and what is defined as protected from ulceration, and how these images were quantified. The PH image looks nothing like the NC image, and yet the quantification indicates these are nearly the same. Please assist the reader in identifying normal mucosa versus regions of ulceration in these images.

Response 9: Thank you for your suggestion. The rats pretreated with LAN and piperine, while the mucosal hemorrhage range was reduced and the cell arrangement tended to be compact and orderly. Among them, the cell arrangement in the PH group was nearly like the NC image. We annotated regions in Figure 3B-C.

Point 10:Figure 4 – Please describe more carefully how these samples were obtained. ROS levels change dramatically upon death of the animal and dissection of the tissue. Were these mucosal samples, full-thickness gastric tissue, or blood? These details should be presented in both the main text as well as the figure legend.

Response 10: Thank you for your suggestion. The sample was collected from ull-thickness gastric tissue. According to your suggestion, we describe the acquisition method of samples in section 2.4.4 and Figure 4 (line 288).

Point 11:Figure 5 – were these tissues collected from full-thickness gastric tissue or only mucosa?

Response 11: Thank you for your suggestion. The tissues were collected from full-thickness gastric tissue in line 158 and 293.

Point 12:Figure 6 – were entire stomachs, including ulcerated tissues, included for protein collection? Please include this information in the methods section.

Response 12: Thank you for your suggestion. The tissues were collected from entire stomachs, including ulcerated tissues, included for protein collection in line 146, 171, 313.

Point 13:For all in vivo studies – while the graphs depict a trend toward best results at the higher dose of piperine, is there a significant difference between the three doses tested? For translation into humans, the lowest effective dose will be important to ascertain.

Response 13: Thank you for your suggestion. There a significant difference between the three doses tested in some indicators (the mRNA and protein expression levels of Nrf2, HO-1, Keap1, ERK, JNK and p38, Figure 5 and 6 ). The lowest effective dose requires further experimental validation.

Point 14:Many of the details pertinent to the study are included in the discussion when they would be better suited to the introduction (description of antioxidants and effects of ethanol on the stomach).

Response 14: Thank you for your suggestion. According to your suggestion, we add some discussion to describe the antioxidants and effects of ethanol on the stomach in line 365, 381, 382, 394-396, 402, 403.

Point 15: Line 325-328 – there was no data describing lipid peroxidation in this manuscript. Oxidative stress can occur from numerous other sources and this study did not determine the cause.

Response 15: Response 14: Thank you for your suggestion. According to your suggestion,we delete the statement of lipid peroxidation in line 336, 338, 341.
